# Bacteria, Fungi, and Protists Exhibit Distinct Responses to Managed Vegetation Restoration in the Karst Region

**DOI:** 10.3390/microorganisms12061074

**Published:** 2024-05-26

**Authors:** Can Xiao, Dan Xiao, Mingming Sun, Kelin Wang

**Affiliations:** 1College of Environment and Ecology, Hunan Agricultural University, Changsha 410128, China; 2Key Laboratory of Agro-Ecological Processes in Subtropical Region, Institute of Subtropical Agriculture, Chinese Academy of Sciences, Changsha 410125, China; smm23@mails.ucas.ac.cn; 3Huanjiang Agriculture Ecosystem Observation and Research Station of Guangxi, Guangxi Key Laboratory of Karst Ecological Processes and Services, Huanjiang Observation and Research Station for Karst Ecosystems, Chinese Academy of Sciences, Huanjiang 547100, China; 4University of Chinese Academy of Sciences, Beijing 100039, China

**Keywords:** bacteria, fungi, protists, karst ecosystems, vegetation restoration

## Abstract

Bacteria, fungi, and protists occupy a pivotal position in maintaining soil ecology. Despite limited knowledge on their responses to managed vegetation restoration strategies in karst regions, we aimed to study the essential microbial communities involved in the process of vegetation restoration. We compared microbial characteristics in four land use types: planted forests (PF), forage grass (FG), a mixture of plantation forest and forage grass (FF), and cropland (CR) as a reference. Our findings revealed that the richness of bacteria and protists was higher in FF compared to PF, while fungal richness was lower in both PF and FF than in CR. Additionally, the bacterial Shannon index in FF was higher than that in CR and PF, while the fungal and protist Shannon indices were similar across all four land use types. Significant differences were observed in the compositions of bacterial, fungal, and protist communities between FF and the other three land use types, whereas bacterial, fungal, and protist communities were relatively similar in PF and FG. In FF, the relative abundance of bacterial taxa Acidobacteria, Firmicutes, and Gemmatimonadetes was significantly higher than in PF and CR. Fungal communities were dominated by Ascomycota and Basidiomycota, with the relative abundance of Ascomycota significantly higher in FF compared to other land use types. Regarding protistan taxa, the relative abundance of Chlorophyta was higher in FF compared to CR, PF, and FG, while the relative abundance of Apicomplexa was higher in CR compared to FF. Importantly, ammonium nitrogen, total phosphorus, and microbial biomass nitrogen were identified as key soil properties predicting changes in the diversity of bacteria, fungi, and protists. Our results suggest that the microbial community under FF exhibits greater sensitivity to vegetation restoration compared to PF and FG. This sensitivity may stem from differences in soil properties, the formation of biological crusts and root systems, and management activities, resulting in variations in bacterial, fungal, and protist diversity and taxa in PF. As a result, employing a combination restoration strategy involving plantation forest and forage grass proves to be an effective approach to enhance the microbial community and thereby improve ecosystem functionality in ecologically fragile areas.

## 1. Introduction

Bacteria, fungi, and protists play an important role in soil ecosystems, with their interactions being crucial for maintaining soil biodiversity [1]. For instance, protists prey upon other microorganisms, establishing complex food chain [2]. These microbial communities, through their mutual relationships, decompose organic matter more efficiently, releasing vital nutrients for plant absorption [3]. Consequently, these microorganisms significantly contribute to enhancing soil nutrient availability, increasing plant productivity, and sustaining ecological balance [4,5].

The diversity and composition of bacterial, fungal, and protistan communities can be significantly influenced by different land use types [6]. Variations in litter and fertilizer input, organic matter decomposition, and plant community structure can all alter soil nutrient availability, thereby regulating microbial diversity and activity. For instance, vegetation restoration from cropland, particularly when it involves diverse plant species and sustainable management practices, has been shown in numerous studies to increase soil carbon (C) and nitrogen (N) stocks [7,8,9,10]. However, there are also instances where vegetation restoration projects have led to a decrease in soil mesoaggregate-associated organic C stocks [11], while soil C and N stocks have remained unchanged or even decreased following afforestation with pine [12]. These conflicting results suggest that the accumulation of soil nutrients following cropland conversion is highly uncertain and may depend on the type and duration of the restoration strategy [13]. Additionally, compared to vegetation restoration, the application of organic and inorganic fertilizers in croplands increases soil nutrient levels, such as available phosphorus (P) [14]. Furthermore, different vegetation restoration strategies, coupled with variations in plant root biomass and root exudates, further influence the distribution of microbial communities [15,16,17,18].

Bacteria, fungi, and protists demonstrate distinct advantages in various nutrient environments. Typically, microbial biomass and activity are positively associated with soil organic matter, as it serves as a source of carbohydrates for microbial growth [19,20]. However, N-enriched conditions can decrease the relative abundance of certain bacterial taxa (e.g., Actinobacteria and Nitrospirae) [21], while increasing the abundance of Ascomycota fungi [22]. Previous studies suggest that fungi, compared to bacteria, exhibit greater adaptability to N-limited environments, indicating distinct response mechanisms to varying nutrient levels [23]. Plant woody debris and fallen leaves are typically preferentially decomposed by fungi (e.g., saprotrophic Ascomycota and Basidiomycota), as they possess strong capabilities for lignin degradation [24,25,26]. In this context, fungi thrive under vegetation restoration with planted trees because of the higher input of plant litter compared to cropland. Conversely, the application of fertilizers in cropland favors bacterial growth over fungi, resulting in a bacterial-dominated food web [27]. Predatory protists act as primary consumers of fungi and bacteria, and any changes in the abundance and diversity of bacteria and fungi further influence the prey availability for protists [28,29,30]. Therefore, the correlation among bacterial, fungal, and protistan taxa can be significantly influenced by different land use types.

Karst ecosystems are widespread in southwestern China and are renowned for their unique geological characteristics [31]. Karst areas are characterized by a thin soil layer, with karst erosion leading to high permeability of bedrock and limited water storage capacity [32]. Over the past century, the rapid growth of the population has led to the extensive conversion of land for agricultural purposes, causing significant land degradation and desertification [33,34]. To address these pressing issues, various vegetation restoration projects have been implemented in degraded karst ecosystems. These include artificial forestation, natural grassland restoration, and forest–grass mixed planting strategies [35,36]. These efforts aim to restore the ecological balance and enhance soil fertility in these fragile environments. Several studies have highlighted the changes in soil microbial functioning in karst environments, particularly concerning plant communities and soil properties [37,38]. However, most of these studies have primarily focused on bacteria and fungi [39,40], leaving a significant knowledge gap regarding protists and their interactive relationships with other soil microorganisms. Given the crucial role of protists in soil ecosystems as primary consumers of bacteria and fungi, it is imperative to fill this gap, especially in the context of managed vegetation restoration strategies [41]. Consequently, there is a need for more detailed knowledge to elucidate the effects of different managed vegetation restoration strategies on bacteria, fungi, and protists in the karst region.

To explore the pivotal microbial communities involved in managed vegetation restoration, we conducted a study on the diversity and community composition of bacteria, fungi, and protists in the karst region. This study delved into three managed vegetation restoration strategies: plantation forest (PF), forage grass (FG), and a mixed approach combining both plantation forest and forage grass (FF). To provide a baseline comparison, cropland (CR) was employed as a control. Our aim was to investigate the characteristics of bacteria, fungi, and protists during vegetation restoration. We hypothesized that a mixed approach combining both plantation forest and forage grass would exert a greater influence on microbial diversity and community composition, as compared to other restoration strategies, by enhancing soil nutrients and promoting a more diverse plant community.

## 2. Materials and Methods

### 2.1. Study Area

The experimental area is situated in the Guzhou catchment (24°54′–24°55′ N, 107°56′–107°57′ E) within Huanjiang County, Guangxi Zhuang Autonomous Region, Southwest China. This region experiences a subtropical monsoon climate, characterized by an average annual temperature and average annual precipitation of 16.9 °C and 1675 mm, respectively. The rainy season primarily occurs from April to September. The Guzhou catchment covers an area of 10 km^2^. The parent rock is limestone, and the soil type is calcareous. The study area is representative of a typical karst landscape, featuring a gently sloping depression at its center surrounded by clusters of steep hills. Bedrock outcrops are high on the slopes compared to the peak-cluster depression area. Soil depth ranges from 50 to 80 cm in the peak-cluster depression to 10 to 30 cm on the slopes.

Before the 1980s, intense deforestation and cultivation accelerated soil and nutrient loss, leading to land degradation and severe desertification in the area. To address these issues, large-scale ecological projects such as returning farmland to forests and implementing ecological migration were initiated for ecological restoration. Artificial economic forests have been established to replace traditional farmland, particularly in sloped positions where maize–soybean rotation was common. Many sloping farmlands have been naturally replaced by planting grass and economic trees. For this study, a total of 16 plots were established (four treatments × four replication) from 2004, with each plot measuring 20 m × 20 m. The treatments included forage grass (FG), planted forests (PF), a mixture of plantation forest and forage grass (FF), and cropland (CR) as a reference (Figure 1).

### 2.2. Soil Sampling

Soil sampling for this study was conducted in August 2022. At each plot, eight sampling points were randomly selected after removing the surface stubble and litter layer. Subsequently, soil samples from these eight points were combined to create one composite sample per plot. The soil samples collected from each plot were sieved through a 2 mm sieve to remove any coarse debris, plant roots, or stones.

The soil samples were divided into three parts for different analytical purposes. One portion of the soil samples was allowed to air dry naturally. These dried samples were used to determine the physical and chemical properties of the soil. Another portion of the soil samples was stored at 4 °C for microbial biomass determination. A third portion of the soil samples was stored in a refrigerator at −80 °C for DNA extraction.

### 2.3. Experimental Analysis Methods

#### 2.3.1. Soil Properties Analysis

Soil pH was measured using a 1:2.5 soil-to-water ratio via a pH meter (meterFE20K; Mettler-Toledo, Greifensee, Switzerland). Soil organic C (SOC) was determined using the dichromate redox colorimetric method. Total N (TN) in the soil was analyzed using an automatic elemental analyzer. Total P (TP) content in the soil was determined using the method of ascorbic acid molybdate. Available P (AP) was measured using the molybdenum blue method. Total K (TK) content in the soil was determined by melting sodium hydroxide and the flame photometric method. Available K (AK) content in the soil was determined by ammonium acetate leaching and the flame photometric method. Exchangeable calcium (Ca^2+^) and magnesium (Mg^2+^) in the soil were determined using inductively coupled plasma optical emission spectrometry (ICP-OES) after ammonium acetate exchange (Agilent, Santa Clara, CA, USA). Soil microbial biomass C (MBC) and microbial biomass N (MBN) were determined using the chloroform fumigation extraction method. Ammonium (NH_4_^+^) and nitrate (NO_3_^−^) in the soil were determined by KCl extraction procedures. Soil dissolved organic C (DOC) was extracted using a solution of 0.5 M K_2_SO_4_ with a soil-to-solution ratio of 1:5 (*w*/*v*) and analyzed using a total organic C analyzer. Soil dissolved organic N (DON) was calculated indirectly by subtracting the amount of mineral N (NH_4_^+^ and NO_3_^−^) from the total dissolved N in the soil extract [42,43].

#### 2.3.2. Soil DNA Extraction

To extract total soil DNA, 0.3 g of soil samples, stored at −80 °C, were weighed using the FAST DNA spin kit for soil (MP Biomedicals, Eschwege, Germany), following the specific procedure outlined in the kit’s instruction manual. The quality of the genomic DNA was assessed through agarose gel electrophoresis (1%), while the concentration and purity of the extracted DNA were determined using an ultraviolet spectrophotometer (NanoDrop2000, Thermo Fisher Scientific, Waltham, MA, USA). 

#### 2.3.3. Amplicon Sequencing and Sequence Analysis

The primer pairs used for amplifying the bacterial 16S rDNA gene were 338F (ACTCCTACGGGAGGCAGCAG) and 806R (GGACTACHVGGGTWTCTAAT). For the amplification of fungal ITS genes, the primer set ITS1F (CTTGGTCATTTAGAGGAAGTAA) and ITS2R (GCTGCGTTCTTCATCGATGC) was employed [44]. For the amplification of protist 18S rRNA genes, the primer set (CCAGCA(G/C)C(C/T) GCGGTAATTCC) and TAReukREV3 (ACTTTCGTTCTTGAT (C/T; A/G)A) was used [45].

The PCR reaction systems comprising a 20 μL mixture containing 10 ng of DNA template, 0.8 μL of each primer (5 μM), 2 μL of 2.5 mM dNTPs, 4 μL of 5× FastPfu buffer, 0.4 μL of TaKaRa rTaq DNA polymerase, 0.2 μL of BSA, and sterile water to adjust the volume. For the amplification of bacterial 16S rDNA gene, the PCR conditions were an initial denaturation at 95 °C for 3 min, followed by 27 cycles of denaturation at 95 °C for 30 s, annealing at 55 °C for 30 s, extension at 72 °C for 45 s, and a final extension at 72 °C for 10 min. For fungal ITS, the PCR program involved an initial denaturation at 95 °C for 3 min, followed by 35 cycles of denaturation at 95 °C for 30 s, annealing at 55 °C for 30 s, extension at 72 °C for 45 s, and a final extension at 72 °C for 10 min. For the protist 18S rRNA gene, the PCR conditions were as follows: an initial denaturation step at 95 °C for 5 min, followed by 10 cycles of denaturation at 94 °C for 30 s, annealing at 57 °C for 45 s, and extension at 72 °C for 60 s. This was followed by 25 cycles of denaturation at 94 °C for 30 s, annealing at 45, 47, 48, and 49 °C for 45 s each, and extension at 72 °C for 60 s. The reaction concluded with a final elongation step at 72 °C for 2 min. To mitigate PCR biases at the reaction level, the three PCR products obtained from each sample were pooled. The PCR products were sent to Shanghai Meiji Biomedical Technology Co. for high-throughput sequencing.

The QIIME v2–2020.2 environment was employed for conducting bioinformatics analyses on all sequence read data. In particular, the raw sequences were denoised to identify amplicon sequence variants (ASVs) using the “data2” plugin. For bacteria, fungi, and protists, the parameter was set to 420, 280, and 400 base pairs (bp), respectively. Following that, reference sequences from SILVA (version 132) were used for bacteria, UNITE (version 9.0) for fungi, and PR2 (version 4.14.0) for protists to train the classifier, ensuring precise taxonomic assignment to the ASVs. After denoising and taxonomic assignment, the ASV table was rarefied using the “rrarefy” function from the “vegan” package (version 2.5.7) in R. In this study, the number of reads obtained from bacteria ranged from 37,842 to 82,776, with an average of 67,366. For fungi, the number of reads ranged from 73,523 to 163,087, with an average of 101,600. Additionally, the number of reads obtained from protists ranged from 18,114 to 64,086, with an average of 39,982. The rarefaction process entailed normalizing the sequence counts to the same sequencing depth, utilizing the lowest number of obtained reads as a reference, through the “rrarefy” function from the vegan (v2.6-4) R package. This ensured a consistent basis for comparison when computing the microbial diversity index. In this study, microbial diversity, including richness and the Shannon index, was calculated using the “RAM v1.2.1.7” R package. More detail about the sequence analysis can be found in our previous studies [46,47].

### 2.4. Statistical Analysis

The present study examined the effect of land use types on the community composition, richness, and the Shannon index of bacteria, fungi, and protists. The differences in soil properties and the diversity of bacteria, fungi, and protists were assessed using ANOVA with Duncan’s post hoc test (with a significance level set at *p* < 0.05). To visualize the dissimilarities among bacterial, fungal, and protist communities based on ASVs, we employed non-metric multidimensional scaling (NMDS) using the “metaMDS” function from the vegan (v2.6-4) R package, utilizing the Bray–Curtis distance etric. Two axes were generated and visualized using the ggplot2 (v3.4.2) R package. Subsequently, we conducted a similarities analysis (ANOSIM) to assess significant differences in the compositions of bacterial, fungal, and protistan communities based on ASVs within pairwise treatments. A two-way ANOVA and permutation multivariate analysis of variance (PERMANOVA) were conducted using the R function “aov” to assess the impacts of economic tree species, planted grass, and their interactions on bacterial, fungal, and protistan diversity, as well as community composition. Additionally, Pearson’s correlation analysis was conducted to assess the relationships between the bacterial, fungal, and protistan taxa, revealing patterns of association. Furthermore, a random forest model was employed to identify the relative importance of soil properties in explaining changes in microbial diversity. The entire data processing was conducted using R version 3.6.3 (R Core Team).

## 3. Results

### 3.1. Soil Physicochemical Properties

The contents of TP, AP, AK, NO_3_^−^, and Mg^2+^ were significantly influenced by vegetation restoration efforts. Specifically, TP, AP, AK, NO_3_^−^, and Mg^2+^ levels in the CR were higher than those observed in the three vegetation restoration types. Conversely, the NH_4_^+^ was lower in CR compared to PF, FG, and FF. There were no significant differences in TP, NH_4_^+^, AP, and AK levels between PF, FG, and FF. The DON, MBC, and MBN were significantly higher in PF compared to CR and FG (Appendix A).

### 3.2. The Diversity and Community Compositions of Bacteria, Fungi, and Protist

The richness of bacteria and protist is higher in FF compared to PF. Fungal richness was higher in CR than in PF and FF. Furthermore, the bacterial Shannon index was higher in FF compared to CR and PF. Similar levels of fungal and protistan Shannon index were observed across the four land use types (Figure 2).

Compared to the diversity of bacteria, fungi, and protists, their community composition was more influenced by the interactions between economic tree species and planted grass (Table 1). The non-metric multidimensional scaling (NMDS) ordination revealed variations in the community composition of bacteria, fungi, and protist across different land use types. Notably, according to the similarities analysis, we observed that the bacterial, fungal, and protistan communities in FF exhibited a distinct composition compared to other vegetation restoration types of PF, FG, and FF. Conversely, the bacterial, fungal, and protistan communities showed relatively similar compositions among the CR, PF, and FG (Figure 3).

The bacterial community composition at the phylum level was primarily dominated by Actinobacteria, Proteobacteria, and Acidobacteria. The relative abundance of Acidobacteria, Chloroflexi, Firmicutes, and Gemmatimonadetes was higher in FF compared to PF and CR, with the exception of Chloroflexi in PF. Conversely, the relative abundance of Proteobacteria and unclassified taxa was lower in FF compared to the other three land use types (Figure 4). At the genus level, *Bacillus*, *Bradyrhizobium*, *Gaiella*, and *Pedomicrobium* were among the bacterial taxa with the highest relative abundance, although a significant portion of bacteria belonged to unclassified taxa. The relative abundance of *Bacillus* and *Bradyrhizobium* was notably higher in FF compared to CR and PF (Appendix A). The fungal community was dominated by Ascomycota and Basidiomycota. The relative abundance of Ascomycota was higher in FF compared to the other land use types, while the trend for unclassified taxa was opposite. Basidiomycota was more abundant in PF than in CR and FF. The relative abundance of Mucoromycota was more prevalent in FF compared to PF and FG (Figure 4). At the genus level, *Archaeorhizomyces*, *Mortierella*, and *Trichoderma* were the dominant taxa for fungi. The *Mortierella* exhibited a higher relative abundance in the FF compared to the CR, PF, and FG, while *Archaeorhizomyces* showed the opposite trend (Appendix A). The main protist phyla were Chlorophyta, Conosa, and Cercozoa. The relative abundance of Apicomplexa was higher in CR than in FF. The relative abundance of Chlorophyta was higher in FF than in CR, PF, and FG. In contrast, the relative abundance of unclassified taxa was higher in CR, PF, and FG compared to FF (Figure 4). At the genus level of protists, *Acramoeba* and *Filamoeba* were the prominent taxa. The relative abundance of Acramoeba was lower in FF compared to PF and FG (Appendix A).

### 3.3. Relationships among Bacterial, Fungal, and Protistan Taxa

Pearson’s correlation coefficients revealed relationships among bacterial, fungal, and protistan taxa. The relative abundance of the protistan phylum Apicomplexa was negatively correlated with the bacterial phylum Firmicutes. The relative abundance of the protistan phylum Chlorophyta was positively correlated with the bacterial phyla Chloroflexi and Firmicutes, while negatively correlated with the fungal phylum Basidiomycota. Additionally, the relative abundance of the protistan phylum Ochrophyta was positively correlated with the fungal phyla Ascomycota and Mucoromycota but negatively correlated with Basidiomycota (Figure 5). Moreover, at the genus level, the predatory protozoans *Acramoeba* and *Filamoeba* exhibit a negative correlation with fungal groups such as *Cyberlindnera*, while demonstrating a positive correlation with unclassified fungal taxa. Additionally, the predatory protozoan *Platyophrya* shows a negative correlation with bacterial taxa like *Bacillus* and fungal taxa like *Trichoderma* (Appendix A).

### 3.4. Key Factors Regulating Microbial Diversity

Among these environmental factors, NH_4_^+^, MBN, and DON were identified as the primary predictors of variations in bacterial richness. NH_4_^+^ and MBN emerged as the strongest predictors of the bacterial Shannon index. TP and AK were highlighted as the most important significant predictors of fungal richness, while DON, SOC, TP, Ca^2+^, and MBC were identified as the most important significant predictors of the fungal Shannon index. Additionally, DOC, TK, SOC, and TN emerged as the strongest predictors of protistan richness, while pH, MBN, and MBC were identified as the most important significant predictors of the protistan Shannon index (Figure 6).

## 4. Discussion

### 4.1. Effects of Vegetation Restoration on the Diversity of Bacteria, Fungi, and Protist

Vegetation restoration is an effective strategy for improving soil nutrient availability by fostering litter decomposition and enhancing the input of root exudates [48]. It is widely recognized that alterations in soil microbial communities are intimately linked to changes in plant community structure and soil properties [49]. Our findings provide insights into the impact of vegetation restoration strategies on soil bacterial communities in karst regions. Specifically, our results demonstrate that bacterial richness and Shannon index are highest in FF, significantly surpassing those of PF. The mixed forests of trees and grasses found in FF provide a more diverse array of resources and ecological niches for bacteria, enhancing bacterial diversity compared to the monoculture plantations in PF [50]. Moreover, the presence of more plant roots in FF compared to other land use types helps mitigate soil erosion and nutrient loss, thereby preserving soil quality and resulting in an increase in soil bacterial richness and Shannon index in FF [9,51].

Fungi play a crucial role in plant detritus decomposition [52]. Previous studies have reported a positive correlation between plant detritus production and fungal diversity [53,54]. Many studies have observed higher fungal diversity in vegetated areas compared to agricultural lands [55]. However, there are also studies that present contrasting findings. For instance, converting rainforests into rubber plantations led to an increase in the abundance of mycorrhizal fungi, accompanied by a rise in bacterial abundance as well [56]. These inconsistent results suggest that the impact of land use types on fungal diversity may vary depending on specific soil conditions and vegetation cover. In our study, we observed higher fungal richness in CR compared to PF and FF. This could be attributed to the stimulation of fast-growing opportunistic fungi by fertilizer addition, while slow-growing fungal decomposers may not be similarly affected [57]. Additionally, previous research has shown that soil nutrient availability, such as AP, has a positive impact on soil fungal diversity. Therefore, the higher soil nutrient availability (e.g., AP) observed in CR compared to vegetation restoration areas could contribute to the increased fungal diversity in croplands.

The diversity trends observed among protists align closely with those of bacteria, with both exhibiting the highest levels in FF. This similarity is likely due to the intricate relationships between certain taxa of protists and bacteria. According to research, the protist diversity index exhibits a significant correlation with bacterial diversity values, but not with the fungal diversity index [58]. In most ecosystems, protists numerically dominate as predators, directly feeding on bacteria for growth and utilizing substances produced by bacteria as a source of nourishment [59,60]; for instance, predatory Cercozoa and Lobosa species prey on Acidobacteria, Proteobacteria, and Ascomycota [61,62]. Therefore, the enhanced bacterial diversity observed in FF likely contributes to the higher protistan richness. Additionally, root biomass, a major source of soil organic matter, can support higher microbial biomass [63]. The larger root biomass in FF likely results in enhanced protistan richness compared to PF [9]. Moreover, soil protist biodiversity is influenced by the complex and dynamic distribution of water in soil pores in a heterogeneous environment [64]. In FF, the mixed landscape of plantation forest and forage grass not only enhances root biomass but also stabilizes soil water, fostering stronger predatory relationships between bacteria and protists. Coupled with the increased soil availability of C sources and water stabilization, these factors collectively explain the higher protist richness in FF compared to PF.

### 4.2. Effects of Vegetation Restoration on Community Composition

Previous studies have demonstrated that vegetation restoration positively influences soil microbial community composition by fostering a nutrient-rich environment for soil microorganisms [65,66]. As hypothesized, the interactions between economic tree species and planted grass significantly affect bacterial, fungal, and protistan community composition (Table 1). The composition of bacterial, fungal, and protistan communities in the mixed plantation forest and the forage grass (FF) was notably distinct from the other three land use types (Figure 3). For example, the relative abundance of Acidobacteria, Chloroflexi, and Firmicutes at the phylum level, as well as *Bacillus* and *Nocardioides* at the genus level, were higher in the FF compared to the CR. This suggests that these taxa thrive in FF possibly due to the increased availability of organic matter sources. The current study identified Acidobacteria, Actinobacteria, Proteobacteria, and Chloroflexi as the dominant soil bacterial phyla across the four land use types. This finding agrees with previous research indicating that Ascomycetes and Actinobacteria are the most abundant phyla in the soil bacterial community due to their strong competitive advantage in carbon utilization [67].

Ascomycota and Basidiomycota emerge as the dominant phyla of soil fungi across all four land use types, consistent with findings from the karst region of Guizhou [68]. This similarity may be attributed to the calcium-rich environment of karst areas, as Ascomycota and Basidiomycota require higher calcium levels for survival compared to other fungi, making them well suited for thriving in such conditions [43]. Meanwhile, Cercozoa, Chlorophyta, and Apicomplexa dominate the soil protist communities across the diverse land use categories. Cercozoa, renowned for their adaptability to varying environmental conditions, remain unaffected by key soil parameters, ensuring their survival in a wide range of soil types [69,70]. Terrestrial Chlorophyta species, adapted to thrive in moist environments, account for their higher relative abundance in FF compared to CR, PF, and FG, where forest–grass mixes contribute to greater root biomass, stabilizing soil moisture levels. However, the relative abundance of Apicomplexa and unclassified taxa is lower in FF compared to other land use types, indicating that these groups may have a competitive advantage in single planting systems rather than mixed ecosystems.

Assessing the intricate interactions between protists, bacteria, and fungi is paramount for gaining insights into the stability and sustainability of ecosystems. Our findings reveal significant correlations among certain protistan taxa and bacterial and fungal taxa. Notably, predatory protists, such as Apicomplexa and Lobosa, exhibited negative correlations with bacterial taxa belonging to Firmicutes and Actinobacteria. Moreover, at the genus level, there was a negative correlation between the predatory protozoan *Platyophrya* and bacterial taxa of *Bacillus*. This observation is supported by previous research suggesting a close association between protist community structure and the bacterial community [58]. Consequently, these results indicate a strong competitive interaction between predatory protozoa and bacteria. Additionally, in our study, Chlorophyta showed a positive correlation with Chloroflexi. This could be attributed to the formation of a microalgal-bacterial symbiotic system, where substances produced by the life processes of microalgae and bacteria, such as oxygen and carbon dioxide, can be utilized by each other [71,72]. Previous research has found that Chlorophyta have an affinity for Firmicutes in microalgal-bacterial experiments [73]. This positive correlation between Chlorophyta and the bacterial taxa of Chloroflexi and Firmicutes can be explained by such symbiotic relationships. It is evident that the interactions between protists and internal bacterial and fungal communities play a crucial role in maintaining soil nutrient functions and promoting vegetation restoration.

### 4.3. Key Factors Affecting the Community of Bacteria, Fungi, and Protists

Many environmental variables can explain the composition of microbial communities [74,75]. The diversity and community composition of bacteria, fungi, and protists differed significantly between CR and FF, but not for PF and FG, suggesting that microbial communities were more sensitive to vegetation restoration under FF compared to PF and FG. In this study, soil properties such as NH_4_^+^, DON, MBN, SOC, and DOC were the main factors regulating changes in the diversity of bacteria, fungi, and protists, indicating that C and N supply play important roles in regulating soil microbial communities. Soil nutrients (e.g., NH_4_^+^, AP, and AK) differed between CR and vegetation restoration, which may be one of the reasons for changes in the diversity and community composition of bacteria, fungi, and protists in FF. A previous study reported that the biomass of biocrust in FF was higher than in other land use types at the same study site [76]. Increased biocrust can improve N fixing, C exudation, and soil water retention capacity [76,77], thereby stimulating more bacterial and protistan species (e.g., N-fixing bacteria). Moreover, planting forage grasses benefits root growth [9]. Thus, the rich root biomass resulting from increased C resource availability in the combined restoration strategy of FF may be another reason for inducing changes in microbial diversity in FF. In contrast, disturbances caused by farmland cultivation and fertilizer addition in the CR treatment may also play a significant role in altering microbial diversity.

Collectively, alterations in soil properties, the establishment of biocrusts and roots, along with management interventions, have significantly influenced the bacterial, fungal, and protist diversity and taxa distribution between FF and CR. Our study highlights the pivotal role of mixed plantations encompassing trees and forage grasses in modulating the diversity and composition of bacteria, fungi, and protists. Nevertheless, there is a lack of recent research on the response of microbial community complexity via co-occurrence network patterns to different managed vegetation restoration strategies. Therefore, future research should aim to investigate these causal relationships in greater depth and on a larger scale.

## 5. Conclusions

Our study emphasizes the potential role of mixed plantations of trees and forage grasses, rather than single plantations of either, in regulating the diversity and composition of bacteria, fungi, and protists. The dominant taxa of bacteria (e.g., Acidobacteria), fungi (e.g., Ascomycota), and protists (e.g., Chlorophyta) were higher in the mixed plantation compared to the plantation forest and cropland rotation. Additionally, the richness of bacteria and protists was highest and significantly greater under mixed tree and forage plantation conditions than under single plantation forest conditions. Our results indicate that forest–grass mixes particularly benefit microbial growth, especially bacteria and protists. This may be achieved by regulating soil properties and the biomass of biological crusts and root systems within the forest–grass mixes. Consequently, mixed plantations of trees and forage grasses represent an effective restoration strategy to improve microbial communities, thereby enhancing nutrient supply during the process of karst vegetation restoration. Future studies should delve deeper into the response of microbial communities to various vegetation restoration management strategies by examining the symbiotic network patterns involved in nutrient cycling.

## Figures and Tables

**Figure 1 microorganisms-12-01074-f001:**
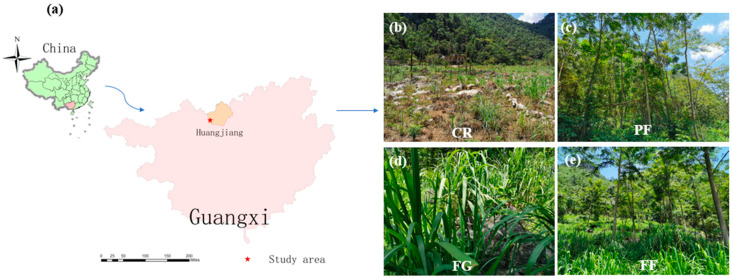
(**a**) Schematic map showing the location of the study area. and the selection of four land-use types, including (**b**) Cropland rotation (CR); (**c**) plantation forest (PF); (**d**) forage grass (FG); and (**e**) plantation forest and forage grass mixed (FF).The FG plots were planted with Guimu-1 hybrid elephant grass, a perennial grass species characterized by its high biomass. This grass can be harvested three to four times annually for use as beef feed. The PF plots were primarily planted with *Zenia insignis*. The leaves of *Z. insignis* can be used as additives for beef feed, while the wood can be utilized for constructing houses, furniture, tools, and other purposes. The FF plots were planted with *Zenia insignis* and Guimu-1 hybrid elephant grass. At the CR plots, maize and soybeans were rotated. Compound fertilizer, comprising both inorganic fertilizer and farm manure, was applied during planting for both maize and soybeans, to enhance their growth and yield.

**Figure 2 microorganisms-12-01074-f002:**
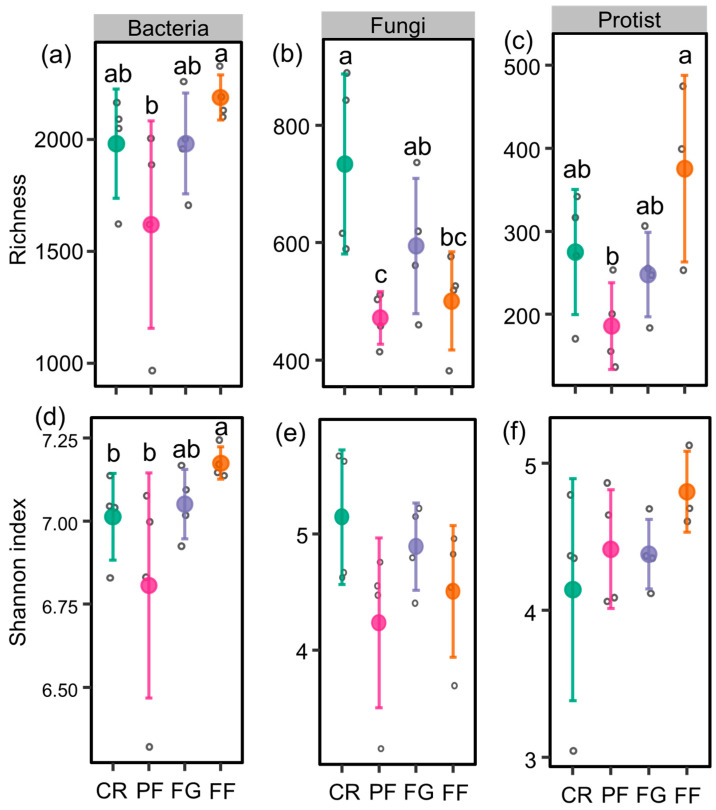
The richness and Shannon index under different vegetation restoration types. (**a**) bacteria richness; (**b**) fungi richness; (**c**) protist richness; (**d**) bacteria Shannon index; (**e**) fungi Shannon index; (**f**) protist Shannon index. CR, PF, FG, and FF represent cropland, planted forests, forage grass, and a mixture of plantation forest and forage grass, respectively. Significant differences (*p* < 0.05) among these vegetation restoration types are denoted by lowercase letters.

**Figure 3 microorganisms-12-01074-f003:**
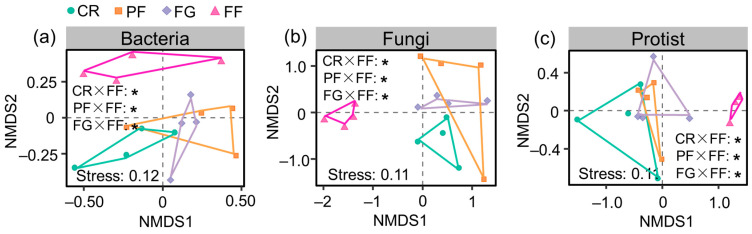
(**a**) Non-metric multidimensional scaling (NMDS) ordinations of bacterial community compositions, (**b**) Non-metric multidimensional scaling (NMDS) ordinations of fungal community compositions, (**c**) Non-metric multidimensional scaling (NMDS) ordinations of protistan community compositions. Significant results of the community composition within pairwise treatments, as determined by the analysis of similarity, are denoted by “*” at *p* < 0.05. CR, PF, FG, and FF represent cropland, planted forests, forage grass, and a mixture of plantation forest and forage grass, respectively.

**Figure 4 microorganisms-12-01074-f004:**
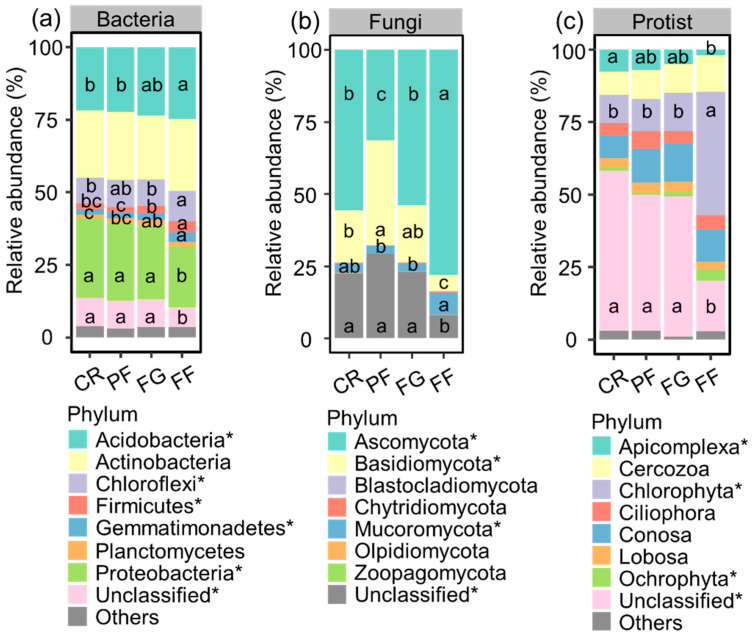
(**a**) Relative abundance of bacteria taxa at the phylum level. (**b**) Relative abundance of fungal taxa at the phylum level. (**c**) Relative abundance of protistan taxa at the phylum level.CR, PF, FG, and FF represent cropland, planted forests, forage grass, and a mixture of plantation forest and forage grass, respectively. The “*” indicates significant differences (*p* < 0.05) among these vegetation restoration types are denoted by lowercase letters.

**Figure 5 microorganisms-12-01074-f005:**
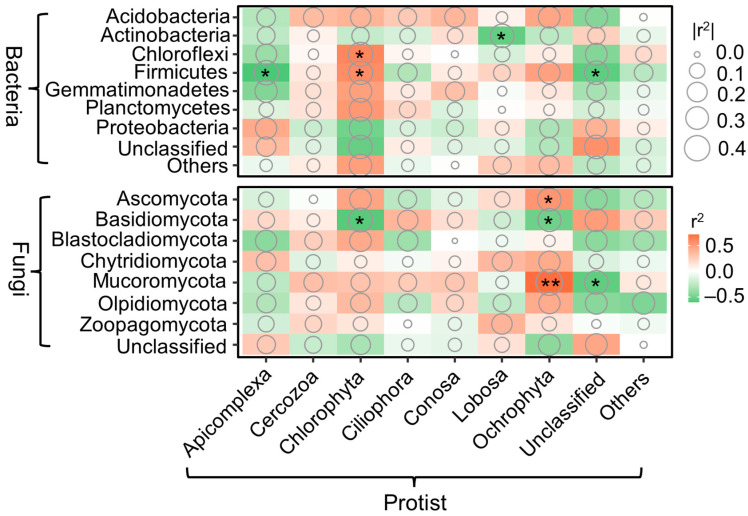
Pearson’s correlation revealing relationships among the phylum-level abundance of bacteria, fungi, and protists. Circle size represents correlation strength, with larger circles indicating stronger correlations. Significant correlation is indicated by asterisks (* *p* < 0.05; ** *p* < 0.01).

**Figure 6 microorganisms-12-01074-f006:**
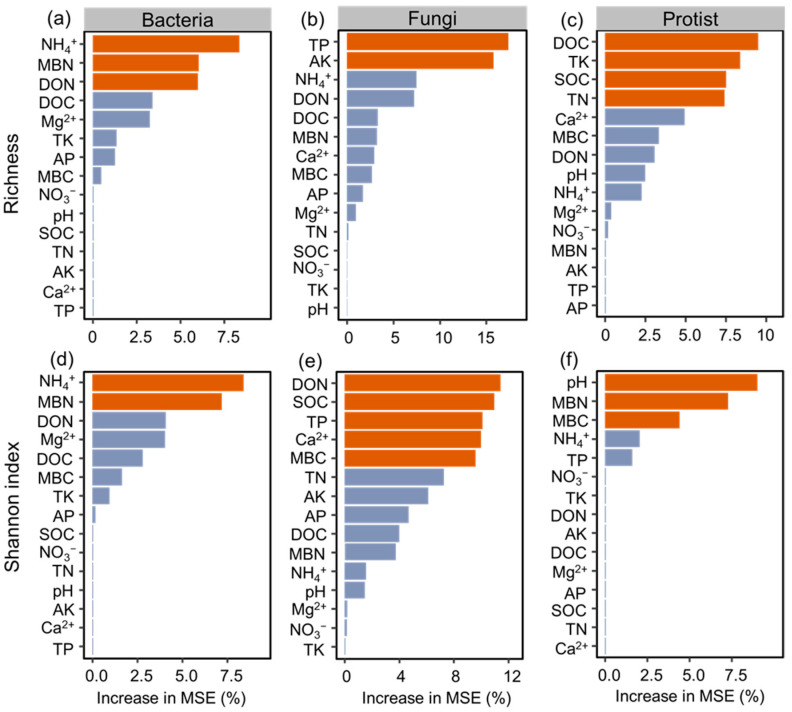
(**a**–**c**) Random forests models showing the effects of soil properties on bacterial, fungal, protistan richness. (**d**–**f**) Random forests models showing the effects of soil properties on bacterial, fungal, protistan Shannon index. The color orange denotes the most important effects.

**Table 1 microorganisms-12-01074-t001:** Effects of economic tree species (Tree), planted grass (Grss), and their interactions on bacterial, fungal, and protistan diversity and community composition.

Items	Bacteria			Fungi			Protist		
	Richness	Shannon Diversity	Composition	Richness	Shannon Diversity	Composition	Richness	Shannon Diversity	Composition
Tree	0.29	0.18	3.13	11.1 *	5.05 **	5.24 *	0.02	1.69	9.19 *
Grass	3.84	4.54 *	2.43	1.06	0.01	6.63 *	3.72	1.57	13.3 *
Tree × Grass	3.84	2.99	5.92 *	2.47	0.83	10.9 *	8.17*	0.09	4.26 *

Data indicate F value, * *p* < 0.05, ** *p* < 0.01.

## Data Availability

The data used to support the findings of this study can be made available by the corresponding author upon request.

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
