# Peer review of "Bacteria, Fungi, and Protists Exhibit Distinct Responses to Managed Vegetation Restoration in the Karst Region"

_microorganisms, 2024, doi:10.3390/microorganisms12061074_

Round 1
Reviewer 1 Report
Comments and Suggestions for Authors
Review of article ID: microorganisms-2999229, titled “Bacteria, Fungi, and Protists Exhibit Distinct Responses to 2 Managed Vegetation Restoration in the Karst Region”.
The article is organized in a logical sequence and with didactic writing that facilitates understanding, even considering the number of microorganisms and their scientific names in the three different microbial communities studied.
I consider it important that the article is subject to careful review by the authors to meet the quality standards of the journal to which they submitted the article. I have some suggestions that could improve the article, overcoming some problems found in the review process. All revisions and recommendations are indicated in the attached text.
Among the recommendations, I argue that the article should replace the coordinates with a map of the location of the studied area, starting from a regional to a local scale, no matter how small the area of the field experiment.
The article table should be cited in the text prior to its insertion.
The figures must be revised in their numbering, as due to the names of some figures in the text, they are not identified in the article.
The materials and methods and the results and discussions are well presented, and the issue of numbering and/or citation of figures should only be resolved.
References must be carefully reviewed and comply with a single referencing standard.
Many references do not indicate the name of the journal of publication.
Some references are incomplete, which does not allow the reader to access them.
Many references do not contain the DOI identifier, which are easily found on the internet.
The conclusions of the article seem more like a continuation of the discussions and should be incorporated into the discussions and the conclusions item should objectively focus on the results obtained and presented throughout the article and their meanings for karst environments.

Author Response
Reviewer 1
Comment: The article is organized in a logical sequence and with didactic writing that facilitates understanding, even considering the number of microorganisms and their scientific names in the three different microbial communities studied.
I consider it important that the article is subject to careful review by the authors to meet the quality standards of the journal to which they submitted the article. I have some suggestions that could improve the article, overcoming some problems found in the review process. All revisions and recommendations are indicated in the attached text.
Author Response: We would like to thank Reviewer 1 for these supportive comments. We have carefully revised the manuscript according to the comments below as well as the attached text.
In addition, we have revised the comments point-by-point from the attached text. Pleased see the revised manuscript.
Overall, we have made every possible effort to revise the manuscript in accordance with the editor’s suggestions above. We hope that these changes have sufficiently enhanced the quality of the manuscript.
Some specific comments
Comment: (1) Among the recommendations, I argue that the article should replace the coordinates with a map of the location of the studied area, starting from a regional to a local scale, no matter how small the area of the field experiment.
Author Response: We appreciate the reviewer's valuable comments. In response, we have added a schematic diagram of the sampling area and sample plot types in Figure 1. This addition provides a clearer general overview of the study area, making it more convenient for readers to understand. Please refer to the revised manuscript, lines 146-148, and Figure 1.
Comment: (2) The article table should be cited in the text prior to its insertion.
Author Response: We thank the reviewer for this suggestion. We have adjusted the placement of the table so that it now appears after it is referenced in the article.
Comment: (3) The figures must be revised in their numbering, as due to the names of some figures in the text, they are not identified in the article. The materials and methods and the results and discussions are well presented, and the issue of numbering and/or citation of figures should only be resolved.
Author Response: We thank the reviewer for this useful suggestion. We have modified the serial numbers of the cited figures to comply with the norms. If the serial numbers appear in the text but the corresponding figures are not visible, please refer to the Supplementary Material.
Comment: (4) References must be carefully reviewed and comply with a single referencing standard. Many references do not indicate the name of the journal of publication. Some references are incomplete, which does not allow the reader to access them. Many references do not contain the DOI identifier, which are easily found on the internet.
Author Response: We thank the reviewer for this suggestion. Based on your comments, we have addressed the issues in the references according to the journal requirements, ensuring that all entries are properly labeled with journal name. Please see the revised References section.
Comment: (5) The conclusions of the article seem more like a continuation of the discussions and should be incorporated into the discussions and the conclusions item should objectively focus on the results obtained and presented throughout the article and their meanings for karst environments.
Author Response: We are grateful for this suggestion. We have rewritten the Conclusions section to focus on the overall findings and summarize the links between vegetation types and soil microbes in karst vegetation restoration strategies. Please see revised manuscript lines 465-479

Reviewer 2 Report
Comments and Suggestions for Authors
Comments on the paper entitled „Bacteria, Fungi, and Protists Exhibit Distinct Responses to Managed Vegetation Restoration in the Karst Region”
General comments
The work concerns the assessment of the diversity and community composition of bacteria, fungi, and protists in the karst region in China after recultivation. The topic undertaken by the authors is important from a cognitive and application point of view, and consequently in the context of environmental protection and recultivation of degraded areas. The manuscript is interesting and well written.
The manuscript may be published after very, very minor revisions.
Specific comments
Line 42: keywords. It is "protist" I propose "protists"
Line 142: What did the fertilizer that was used to fertilize the soil under the crops contain?
Line 222: section 2.4 The authors reported Shannon diversity, Richness and Composition in the Results section. They didn't mention it in the Material and Methodology section. Please complete this information.
Author Response
Reviewer 2
General considerations
Comment: The work concerns the assessment of the diversity and community composition of bacteria, fungi, and protists in the karst region in China after recultivation. The topic undertaken by the authors is important from a cognitive and application point of view, and consequently in the context of environmental protection and recultivation of degraded areas. The manuscript is interesting and well written.
Author Response: We thank reviewer 2 for the supportive comments and helpful suggestions. We have carefully revised the manuscript according to your following comments. We hope that these changes have sufficiently enhanced the quality of the manuscript.
Some specific comments
Comment: (1) Line 42: keywords. It is "protist" I propose "protists".
Author Response: Thank you for the valuable suggestion. We have changed keyword protist to protists. Please see revised manuscript line 43.
Comment: (2) Line 142: What did the fertilizer that was used to fertilize the soil under the crops contain?
Author Response: We appreciate this useful suggestion from the reviewer. We have already explained about compound fertilizers. Compound fertilizer, comprising both inorganic fertilizer and farm manure, was applied during planting for both maize and soybeans to enhance their growth and yield. Please see revised manuscript lines 143-145.
Comment: (3) Line 222: section 2.4 The authors reported Shannon diversity, Richness and Composition in the Results section. They didn't mention it in the Material and Methodology section. Please complete this information.
Author Response: We would like to thank the reviewer for this supportive suggestion. We have added this information in Material Methods section. In this study, microbial diversity, including richness and Shannon index, was calculated using the "RAM v1.2.1.7" R package. The present study examined the effect of land use types on the community composition, richness, and Shannon index of bacteria, fungi, and protists. Please see revised manuscript lines 225-230.
